# Serum Vitamin D Concentrations, Time to Pregnancy, and Pregnancy Outcomes among Preconception Couples: A Cohort Study in Shanghai, China

**DOI:** 10.3390/nu14153058

**Published:** 2022-07-26

**Authors:** Yu Zhang, Anne Marie Z. Jukic, Heqing Song, Lifeng Zhang, Fengyun Yang, Shoule Wu, Dongxiao Yin, Hong Jiang

**Affiliations:** 1School of Public Health, National Health Commission Key Laboratory of Health Technology Assessment, Fudan University, Shanghai 200032, China; 17211020124@fudan.edu.cn (Y.Z.); 20211020051@fudan.edu.cn (H.S.); 2Vital Statistics Department, Songjiang District Center for Disease Control and Prevention, Shanghai 201600, China; 3Epidemiology Branch, National Institute of Environmental Health Sciences, Durham, NC 27709, USA; jukica@niehs.nih.gov; 4Shanghai Jiading Maternal and Child Health Care Hospital, Shanghai 201812, China; 18930862729@163.com (L.Z.); jdyfy@126.com (F.Y.); wushoule@126.com (S.W.); jdfbk2010@163.com (D.Y.)

**Keywords:** vitamin D, 25-hydroxyvitamin D, conception, time to pregnancy, pregnancy outcomes

## Abstract

Background: The role of vitamin D in reproductive health is still unclear. This study aimed to assess the effect of serum 25-hydroxyvitamin D (25(OH)D), among preconception couples, on fecundity, and the associations between 25(OH)D concentrations before and during pregnancy, and pregnancy outcomes. Methods: 200 preconception couples attempting to conceive were recruited and were followed-up until childbirth. Time to pregnancy was collected via telephone every two months or obtained via a questionnaire during pregnancy. Blood samples were collected to measure serum 25(OH)D levels from both partners at enrollment and from women during the second and third trimester of pregnancy. Results: Couples had higher conception rates within six months (adjusted odds ratio (aOR): 3.72, 95% CI: 1.16, 11.9) and reduced time to pregnancy (adjusted fecundability ratio (aFR): 1.50, 95% CI: 1.01, 2.23) if male partners had sufficient 25(OH)D compared with insufficient 25(OH)D. Compared to pregnant women with insufficient 25(OH)D in the third trimester of pregnancy, sufficient 25(OH)D was associated with reduced odds of anemia (OR: 0.22, 95% CI: 0.06, 0.82), longer gestational age (β: 0.53, 95% CI: 0.04, 1.01) and newborns’ higher ponderal index (β: 0.10, 95% CI: 0.01, 0.19). Conclusions: Sufficient serum 25(OH)D levels among preconception men or during pregnancy were associated with better reproductive health.

## 1. Introduction

The classic role of vitamin D is to promote the absorption of calcium and phosphorus to maintain calcium homeostasis and bone mineralization [1]. Accumulating evidence suggests that vitamin D deficiency plays an important role in the development of chronic, infectious, and autoimmune diseases, such as hypertension, diabetes, metabolic syndrome, colorectal cancer, and pulmonary tuberculosis [2,3,4,5]. Biological functions of vitamin D are mediated through vitamin D receptors (VDR) expressed in organs, such as the skin, pancreas, pituitary gland, and skeleton [6]. VDR is also expressed in the reproductive organs, such as ovaries, placenta, and endometrium in females [7,8], and genital tract and testicular tissue in males [9,10]. This implies a potential role for vitamin D in maintaining reproductive health [11].

Vitamin D deficiency and insufficiency has been recognized as a global public health problem that occurs in children, adolescents, pregnant women, and the elderly worldwide [12]. Although there are many studies investigating the role of vitamin D during pregnancy in pregnancy outcomes, the findings are inconsistent [13,14,15]. Most studies have reported low gestational serum 25-hydroxyvitamin D (25(OH)D) linked to an increased risk of pre-eclampsia, gestational diabetes mellitus (GDM), gestational anemia, preterm birth (PTB) and low birth weight (LBW) [16,17,18,19]. Some studies suggested vitamin D deficiency during pregnancy was associated with cesarean section [20,21]. However, very few studies observed the impact of preconception 25(OH)D concentrations on pregnancy outcomes.

Animal studies have shown that vitamin D deficiency is associated with decreased fertility and reduced litter size [22,23,24]. In humans, most studies have investigated the importance of vitamin D in patients undergoing assisted reproductive treatments [25]. However, limited studies have examined vitamin D and fecundity in humans who were not undergoing fertility treatment. Currently, there are four population-based studies that focused on the relationship between 25(OH)D and fecundability. One of the four studies found that 25(OH)D concentrations were not associated with women’s fecundity [26,27]. However, the other three studies found women were more likely to successfully conceive with higher 25(OH)D concentrations [10,27,28]. Furthermore, only one of the above studies measured 25(OH)D concentrations for both preconception females and males but with a relatively small sample size of 132 couples [10]. To date, there have been limited data elucidating the effect of preconception vitamin D concentrations among healthy couples on conception and pregnancy outcomes.

We proposed this study to assess the effect of preconception serum 25(OH)D concentrations among both females and males on the rate of clinical pregnancy and time to pregnancy, and to examine the associations between 25(OH)D concentrations before and during pregnancy, and pregnancy outcomes.

## 2. Materials and Methods

### 2.1. Study Design

This study was a cohort study based on the Preconceptional Offspring Trajectory Study (PLOTS) [29,30]. PLOTS was a prospective cohort study which aimed to investigate the effects of environmental exposure, social determinants of health, lifestyle, and different healthcare service patterns on preconception couples’ fertility, pregnancy outcomes and offspring health. From September 2016 to November 2018, preconception couples who attempted to conceive and attended a pre-pregnancy health checkup at the preconception healthcare clinic of the Maternal and Child Health Hospital of Jiading District, Shanghai, China, were recruited in the study. The inclusion criteria were: (1) attempting to conceive at the time of recruitment; (2) between 20 and 49 years old for the female; (3) no known diagnosis of infertility. The exclusion criteria included a self-reported history of clinical diagnosis of infertility before conception, pelvic inflammatory disease, tubal occlusion, endometriosis, anovulation and polycystic ovarian syndrome, or uterine abnormality. The couples were followed-up from recruitment to a clinically confirmed pregnancy or 12 months of attempting. Participants who conceived were followed throughout pregnancy.

After informed consent was obtained, couples were asked to complete a baseline questionnaire that ascertained demographic and lifestyle information. Participants completed an additional questionnaire in the third trimester of pregnancy to collect health and lifestyle information. Blood samples were collected from both partners of the preconception couple at recruitment and, for women who conceived, in the second and third trimesters of pregnancy. A semen sample was collected at recruitment from male partners. By September 2019, a total of 260 participants in the cohort gave birth in the hospital, 200 of whom had both valid questionnaire information and complete blood samples for each visit. Therefore, a total of 200 preconception couples were included in this study, and 198 women were analyzed during pregnancy after excluding 2 participants who had non-singleton pregnancies. Cohort couples who achieved a clinical pregnancy through in vitro fertilization during the follow-up period were excluded (*n* = 0).

### 2.2. Biochemistry

Both partners of the recruited couple underwent the same blood draw protocol. Fasting blood samples (5 mL) were collected by venipuncture at the time of preconception recruitment and for the female partners who conceived, in the second and third trimesters of pregnancy. After centrifugation at 3000 rpms for 10 min, serum was separated and stored at −80 °C. The quantitative detection of serum 25(OH)D levels was performed by SW and YZ, with MAGLUMI^®^ 2000 at the clinical laboratory of the Maternal and Child Health Hospital of Jiading District of Shanghai using a commercially available kit based on an automated chemiluminescence immunoassay method [31]. The intra- and inter-assay coefficients of variations were below 10% and 15%, respectively. Measurement interval was 3~150 ng/mL. Vitamin D status before and during pregnancy in this study was determined as sufficient with serum 25(OH)D ≥ 30 ng/mL or insufficient with 25(OH)D < 30 ng/mL based on the Endocrine Society guidelines [32]. These cutoffs were applied to pregnancy although there are no recommendations specific for pregnant women.

Semen samples were collected via masturbation on the day they attended a pre-pregnancy health checkup at the preconception healthcare clinic of the research hospital. The total sperm count, sperm concentration, the number of progressive motile sperm and percentage of normal morphology sperm were measured by a sperm quality analyzer [33].

### 2.3. Covariates

Demographic and lifestyle information was collected from both women and men and included their age, education, income, gravidity (only from women), smoking status, alcohol consumption, multivitamin supplementation, calcium supplementation, folic acid supplementation, frequency of milk intake, frequency of animal liver intake, and frequency of deep-sea fish food intake before pregnancy. In addition, body weight and height were measured with a standardized approach by the research investigators. Body mass index (BMI) (kg/m^2^) was calculated for each participant and was categorized as underweight (BMI < 18.5 kg/m^2^), normal weight (18.5 to <24 kg/m^2^), or overweight and obese (≥24 kg/m^2^) according to the recommendation by Working Group on Obesity of China [34].

Weight gain during pregnancy was obtained by calculating the difference between weight measured at delivery and weight measured at the preconception visit. Pregnancy weight gain was categorized into inadequate weight gain, appropriate weight gain, or excessive weight gain according to IOM recommendations [35].

### 2.4. Outcomes

The study outcomes of interest included the rate of conceiving a clinical pregnancy within six months and time to pregnancy (TTP). Clinical pregnancies were confirmed by ultrasound at 6–7 weeks of gestation. TTP was defined as the number of menstrual cycles it took a couple to conceive (and carry the pregnancy to clinical recognition or a live birth) [36,37,38].

In this study, TTP information was collected via both prospective and retrospective methods accounting for 45% (90/200) and 55% (110/200) of couples, respectively. The prospective method included telephone contact with cohort couples every two months to obtain conception information, including contraception use, date of the last menstrual period (LMP), and pregnancy status during the last two months. The following formula was used to calculate total cycles of TTP among participants with prospective data: (LMP date the female partners reported for pregnancy at the most recent tele-follow-up visit − LMP date of baseline questionnaire − days of contraception use during the follow-up period)/cycle length +1 + number of cycles trying to conceive prior to enrollment. The number of cycles was rounded to the nearest whole number. Those who did not respond to telephone follow-ups and did not have prospective TTP information were contacted by study staff at their prenatal care appointments at the research hospital. Retrospective data were collected with a questionnaire that included information on LMP for the pregnancy, duration of contraception since they participated in the cohort, miscarriage after enrollment, etc. The formula to calculate the retrospective TTP was: (LMP date the female partners reported for the pregnancy − LMP date of baseline questionnaire − days of contraception use during the follow-up period)/cycle length +1 + number of cycles trying to conceive prior to enrollment.

After delivery, pregnancy outcomes, including GDM, gestational hypertension, gestational anemia, premature rupture of membranes (PROM), gestational age, birth weight and birth length, were obtained through medical and delivery records review. Pregnant women at 24 weeks of gestation were diagnosed as GDM when their blood glucose levels reached or exceeded the threshold values (fasting: 5.1 mmol/L, 1 h: 10.0 mmol/L, 2 h: 8.5 mmol/L) after an oral glucose tolerance test [39]. Anemia was defined by trimester-specific hemoglobin (Hgb) levels: (1) Hgb levels below 110 g/L in the first trimester; (2) Hgb below 105 g/L in the second trimester; and (3) Hgb below 110 g/L in the third trimester [40]. Pregnancy and birth outcomes included birth weight, and neonatal ponderal index (PI). PI [41,42], as an indicator of fetal nutrition, was calculated using the formula [weight (in grams) ∗ 100]/[length (in centimeters)]^3^. We also explored the association between preconception and gestational serum 25(OH)D concentrations and GDM, gestational anemia, and gestational age at delivery.

### 2.5. Statistical Analysis

Data were analyzed by using the Statistical Package for Social Sciences (SPSS) version 22. The description of continuous variables was expressed by mean/median (M) ± standard deviation ($\bar{x}\pm s$), and quartile (P_25_, P_75_), while categorical variables were expressed by number of cases (*n*) and percentage (%). Chi-square tests were used to analyze the differences between groups of categorical variables, and the independent sample t test, analysis of variance, and non-parametric test were used for the differences of mean values between groups of continuous variables. Spearman correlation coefficient was calculated for serum 25(OH)D levels between men and women.

Multivariate binary logistic regressions were used to analyze the association between serum 25(OH)D levels and the rate of conception within six months of follow-up for women and men, respectively, controlling for preconception age, preconception BMI, education, smoking, alcohol consumption, calcium supplementation, folic acid supplementation, and frequency of milk intake before pregnancy in Model 1, and then further including gravidity, multivitamin supplementation, and season of blood sample collection in Model 2. Cox proportional hazard regression model for discrete survival time was used to evaluate the association between serum 25(OH)D levels and TTP, and to calculate the fecundability ratio (FR) and its 95% CI. The associations between vitamin D and pregnancy and birth outcomes were analyzed after adjusting for potential confounding factors in multivariate linear regression and binary logistic regression model. Statistical significance was defined as *p* < 0.05.

### 2.6. Ethical Approval

This study was approved by Ethics Committee of School of Public Health, Fudan University (IRB00002408 & FWA00002399) and all participants provided written informed consent.

## 3. Results

### 3.1. Descriptive Analysis

The age range of preconception women was 21 to 38 years, with a mean of 28. For preconception men, the range was 22 to 41 years, with a mean of 29. Nearly 70% of the women had at least a college education. About two-thirds of the women were primiparous. Less than a quarter of women reported multivitamin and calcium supplementation before pregnancy and very few men had multivitamin and calcium supplementation (Table 1). Among 200 couples whose baseline preconception 25(OH)D concentrations were measured, the median 25(OH)D concentration of preconception women was 22.42 ng/mL, and the median 25(OH)D concentration of preconception men was 24.17 ng/mL. Concentrations of 25(OH)D from preconception women and men were correlated (r_s_ = 0.368, *p* < 0.001). Compared to women with insufficient preconception 25(OH)D, women with sufficient 25(OH)D were older and more likely to be primiparous and to use a multivitamin or folic acid supplement. Preconception men with older age and using multivitamin, calcium or folic acid supplements were likely to have higher 25(OH)D, and there was a tendency that preconception men with higher BMI to have lower 25(OH)D levels. Blood samples collected in the summer and autumn, compared with winter and spring, were more likely to be sufficient for both women and men. Women or men with a higher BMI were likely to have lower 25(OH)D. Men with higher 25(OH)D had a higher percentage of normal morphology sperm (Table 1).

### 3.2. Probability of Clinical Pregnancy within Six Months

Among 200 preconception couples, there were 151 and 193 confirmed clinical pregnancies within six months and twelve months of follow-up, respectively. Three of them experienced miscarriages after being recruited in the study. In Model 1, after controlling for preconception age, preconception BMI, education, smoking, alcohol consumption, calcium supplementation, folic acid supplementation, and frequency of milk intake before pregnancy, multivariate binary logistic analysis showed that couples were 3.3 times likely to conceive within six months of follow-up if men had sufficient preconception 25(OH)D (≥30 ng/mL), compared to men with insufficient 25(OH)D (<30 ng/mL) (aOR: 3.29, 95% CI: 1.07, 10.12). When further considering gravidity, multivitamin supplementation, and season of blood sample collection in Model 2, the association remained significant with an aOR of 3.72 (95% CI: 1.16, 11.86). When examining serum 25(OH)D among preconception women, there were no significant differences between women whose serum 25(OH)D was sufficient (≥30 ng/mL) or insufficient (<30 ng/mL) in both Model 1 (aOR: 1.02, 95% CI: 0.37, 2.76) and Model 2 (aOR: 0.90, 95% CI: 0.32, 2.54) (Table 2).

### 3.3. Time to Pregnancy

The median time to pregnancy among 200 preconception couples was four (3, 6) menstrual cycles. Cox proportional hazard regression analysis showed that couples had a shorter TTP if men had sufficient preconception 25(OH)D (≥30 ng/mL) compared to those with insufficient 25(OH)D (<30 ng/mL), but without statical significance in Model 1 (aFR: 1.37, 95% CI: 0.93, 2.00). The association altered significantly after further adjusting for gravidity, multivitamin supplementation, and season of blood sample collection in Model 2 (aFR: 1.50, 95% CI: 1.01, 2.23). When compared to women with insufficient 25(OH)D (<30 ng/mL), preconception women with sufficient 25(OH)D (≥30 ng/mL) had a slightly shorter TTP, but the confidence interval was wide both in Model 1(aFR: 1.07, 95% CI: 0.69, 1.68) and Model 2 (aFR: 1.08, 95% CI: 0.70, 1.69) (Table 2).

### 3.4. 25(OH)D during Pregnancy

During the second trimester of pregnancy, the median 25(OH)D concentration of the 198 women was 21.64 ng/mL. The rate of 25(OH)D sufficiency (≥30 ng/mL) and insufficiency (<30 ng/mL) was 14.1% and 85.9%, respectively. During the third trimester, the median 25(OH)D concentration was 19.08 ng/mL. The rate of 25(OH)D sufficiency (≥30 ng/mL) and insufficiency (<30 ng/mL) was 13.1% and 86.9%, respectively. None of the 198 pregnant women were smoking or drinking. Compared with pregnant women with insufficient 25(OH)D, women with sufficient 25(OH)D levels were more likely to have multivitamin and folic acid supplements during the second trimester, and they were more likely to have calcium supplementation and intake of animal livers during the third trimester (*p* < 0.05). Furthermore, pregnant women with sufficient 25(OH)D levels had lower risk of gestational anemia and increased neonatal PI than women with insufficient 25(OH)D levels during the third trimester (*p* < 0.05) (Table 3).

Multivariate binary logistic regression showed that women with sufficient 25(OH)D in the third trimester were negatively correlated with the rate of gestational anemia (OR: 0.22, 95% CI: 0.06~0.82, *p* = 0.02). Multivariate linear regression showed that, compared with the women with insufficient 25(OH)D levels in the third trimester, women with sufficient 25(OH)D level had longer gestational age (β: 0.53, 95% CI: 0.04~1.01, *p* = 0.03) and higher PI among offspring (β: 0.10, 95% CI: 0.01~0.19, *p* = 0.04). No significant statistical associations were found between 25(OH)D levels before pregnancy or during the second trimester of pregnancy and gestational diabetes mellitus, PROM, delivery gestational age, birth weight and neonatal PI (Table 4).

The summary of the findings is shown in Appendix A.

## 4. Discussion

To our knowledge, this is one of few studies in China that has prospectively examined the impacts of serum 25(OH)D levels on fecundity and pregnancy outcomes starting at preconception. Our findings suggest vitamin D deficiency and insufficiency were common among preconception couples. Compared with insufficient 25(OH)D (<30 ng/mL), sufficient 25(OH)D (≥30 ng/mL) among male partners was associated with a higher rate of conception within six months and with shorter TTP. No association was found between serum 25(OH)D levels of preconception women and either the conception rate within six months of follow-up or TTP duration. Further, pregnant women with a high 25(OH)D level (≥30 ng/mL) in the third trimester had increased gestational age at delivery, increased neonatal PI, and reduced risk of gestational anemia.

In this study, there was no association between serum 25(OH)D concentrations in preconception women and conception, which is consistent with one previous study among 203 healthy Danish women [26]. However, three other studies have reported that higher 25(OH)D was associated with increased conception rates. A study of 132 healthy women in the northeast of the US found that women with a 25(OH)D at least 50 nmol/L equivalent to 20 ng/mL had three times the odds of clinical pregnancy within six months [10]. Another study, including 1191 women with a history of pregnancy loss in the USA, reported an increased conception rate with higher 25(OH)D concentrations (aOR: 1.10, 95% CI: 1.01~1.20), but the study also found no association between preconception vitamin D and TTP, which is consistent with our findings [28]. Finally, a recent study of 522 fertile women from central North Carolina found that compared to women with a 25(OH)D of 30–40 ng/mL, women with at least 50 ng/mL had an estimated 35% increase in fecundability (FR: 1.35, 95% CI 0.95–1.91), and women below 20 ng/mL had an estimated 45% reduction in fecundability (FR 0.55, 95% CI 0.23, 1.32). Across three categories (25(OH)D of <20 ng/mL, 30–40 ng/mL and >50 ng/mL), the probability of taking longer than 6 months to conceive was 51%, 28% and 15%, respectively [27]. Animal models and in vitro studies have suggested that low 25(OH)D concentrations were associated with impaired fertility [22,23,43,44,45], while the underlying mechanism of this association remains unclear. Fertilization can be influenced by 1,25(OH)_2_D–VDR mediated nongenomic activity. Besides uterine receptivity and embryonic implantation, HOXA10 expression was upregulated by active vitamin D in an in vitro study of human endometrium, which might also explain the observed associations between vitamin D and fecundity [27,46]. The possible reasons to explain the differences between our study and other studies might be the sample size, vitamin D cutoffs, ethnicity of the study subjects, the study design, and the generally low levels of 25(OH)D in our population, which limited our ability to observe associations at higher 25(OH)D levels. Given the relative paucity of data in this area, more studies of multiple centers with larger sample sizes are needed to explore and confirm the associations between vitamin D and fecundity in the future.

Our results indicated men with a higher percentage of normal morphology sperm were more likely to have higher 25(OH)D levels. Semen quality analysis is an important method to evaluate male fertility. Currently, the effect of vitamin deficiency on semen quality remains controversial. In a cross-sectional study, men with vitamin D deficiency (<25 nmol/L) had a lower proportion of motile, progressive motile and normal morphology sperm (*p* < 0.05) compared with men with high vitamin D concentrations above 75 nmol/L [47]. An animal study reported that fertility of rats was reduced by 73% in litters from vitamin D deficient male inseminations when compared to female rats inseminated by vitamin D replete males [48]. Our study is consistent with these findings: vitamin D may play a role in male fertility.

Very few studies have included preconception men in examining vitamin D concentration and fecundity. In our study, the positive association was found between sufficient 25(OH)D concentrations among preconception men and time to pregnancy. An observational study conducted among 102 infertile couples found that the conception rate was significantly higher among male partners with normal vitamin D serum level (≥30 ng/mL) compared with low vitamin D serum level (<30 ng/mL) [49]. Some studies have suggested the possible mechanism that actives vitamin D could improve sperm–egg binding by increasing intracellular concentrations of calcium, and therefore increased acrosine activity which facilitates fertilization [50,51]. However, randomized clinical trials are needed to further determine whether systemic changes in vitamin D metabolites can influence men’s reproduction.

Our finding that vitamin D sufficiency in the third trimester of pregnancy was associated with a lower rate of gestational anemia agrees with several prior studies, which suggests that maternal vitamin D deficiency in pregnancy may play a role in gestational anemia [18,52]. There are several possible mechanisms to explain the relationship between vitamin D deficiency and anemia. Vitamin D deficiency may upregulate hepcidin—an iron-regulating peptide hormone made in the liver [53], which decreases hemoglobin concentrations and may contribute to anemia [54]. The transcriptional suppression of hepcidin gene (HAMP) expression mediated by 1,25-dihydroxyvitamin D binding to the VDR can cause the lower levels in hepcidin mRNA [55]. Therefore, lower vitamin D levels were associated with higher hepcidin levels, which may reduce the expression of iron transporters in cell membranes, and thus restrict iron transport.

In the current study, vitamin D levels in the third trimester of pregnancy were associated with gestational age at delivery. Our results are consistent with several previous studies. For example, Morley et al. reported that gestational length was 0.7 week shorter (*p* < 0.05) in mothers with low vitamin D levels vs. mothers with higher vitamin D levels [56]. In a multi-ethnic cohort study, pregnant women with higher vitamin D levels (≥20 ng/mL) were reported to have a longer gestational age than women with lower vitamin D levels (<20 ng/mL) [57]. However, there were also studies demonstrating a null relationship between vitamin D level and gestational age of delivery [58].

Our present study showed 25(OH)D sufficiency (≥30 ng/mL) in the third trimester of pregnancy was associated with increased neonatal PI. While several systematic reviews suggested that maternal vitamin D status had no effect on offspring birth length [59,60], there were some studies displaying vitamin D deficiency of pregnant women might have a role on reduced birth weight [13,61]. PI is a more reasonable indicator reflecting physical development and adiposity of infant than length and weight. VDR in placenta plays an important role in pregnancy and maternal VDR gene polymorphism which may affect birth weight [62]. Interestingly, a recent study proposed that maternal vitamin D status during pregnancy may be a significant determinant of the off-spring’s telomere length, which is positively correlated with neonatal weight [63].

Our study was one of few studies examining both preconception women and men with respect to vitamin D and fecundity. The cohort design provided the opportunity to estimate the casual effect of vitamin D on the probability of pregnancy and pregnancy outcomes. The positive association between sufficient vitamin D among preconception men and the increased rate of conception joins a growing body of evidence supporting the importance of preconception vitamin D among fertile couples seeking natural conception, and it provides evidence for updating and improving pre-pregnancy healthcare guidelines. However, we had a relatively small sample size for our analysis of PTB and LBW. In addition, our study only included couples who had a successful pregnancy and live birth. About half of the TTP data in this study were acquired retrospectively during pregnancy, which might lead to misclassification. Moreover, we used 30 ng/mL as sufficiency cutoff point according to the Endocrine Society guidelines [32]. However, it is needed to ascertain the associations between vitamin D and reproductive health outcomes using different cutoff points since there was a debate on whether guidelines derived from studies of North American participants are applicable to participants in other human populations, such as Asian populations.

## 5. Conclusions

Vitamin D deficiency and insufficiency were common among preconception couples. Couples whose male partners had sufficient 25(OH)D (≥30 ng/mL) had a higher rate of conception within six months and a shorter TTP. Women with sufficient 25(OH)D levels during the third trimester of pregnancy had a lower rate of gestational anemia, a later gestational age at delivery, and their infants had an increased neonatal PI. This study suggested that sufficient male 25(OH)D concentrations before conception might have a positive effect on couple’s fecundity, while female 25(OH)D levels were not associated with fecundity. More studies with a larger sample size, a broad distribution of 25(OH)D levels, and both male and female partners, are needed to explore and confirm the associations between vitamin D and fecundity and pregnancy outcomes in the future.

## Figures and Tables

**Table 1 nutrients-14-03058-t001:** Participants’ characteristics by preconception 25(OH)D concentrations in the study (*n* = 200).

Characteristics	Female, *n* (%) or M (P_25_, P_75_)			Male, *n* (%) or M (P_25_, P_75_)		
	Overall	25(OH)D Groups		Overall	25(OH)D Groups	
		<30 ng/mL	≥30 ng/mL		<30 ng/mL	≥30 ng/mL
Serum 25(OH)D concentrations(ng/mL)	22.42 (19.22, 27.48)	173 (86.5)	27 (13.5)	24.17 (21.08, 28.49)	163 (81.5)	37 (18.5)
Age, years						
<28 (female)/29 (male)	95 (47.5)	88 (92.6)	7 (7.4)	95 (47.5)	79 (83.2)	16 (16.8)
≥28 (female)/29 (male)	105 (52.5)	85 (81.0)	20 (19.0)	105 (52.5)	84 (80.0)	21 (20.0)
Body mass index, kg/m^2^						
<18.5	30 (15.0)	22 (73.3)	8 (26.7)	11 (5.5)	11 (100.0)	0 (0.0)
18.5–23.9	150 (75.0)	132 (88.0)	18 (12.0)	94 (47.0)	73 (77.7)	21 (22.3)
≥24.0	20 (10.0)	19 (95.0)	1 (5.0)	95 (47.5)	79 (83.2)	16 (16.8)
Education						
Completed college education or higher	138 (69.0)	117 (84.8)	21 (15.2)	185 (92.5)	151 (81.6)	34 (18.4)
Completed high school education or lower	62 (31.0)	56 (90.3)	6 (9.7)	15 (7.5)	12 (80.0)	3 (20.0)
Household annual income per capita						
≥¥100,000	119 (59.5)	105 (88.2)	14 (11.8)	119 (59.5)	98 (82.4)	21 (17.6)
<¥100,000	81 (30.5)	68 (84.0)	13 (16.0)	81 (30.5)	65 (80.2)	16 (19.8)
Gravidity						
≥1	64 (32.0)	60 (93.8)	4 (6.3)	-	-	-
0	136 (68.0)	113 (83.1)	23 (16.9)	-	-	-
Smoking						
Yes	1 (0.5)	1 (100)	0 (0.0)	48 (24.0)	35 (72.9)	13 (27.1)
No	199 (99.5)	172 (86.4)	27 (13.6)	152 (76.0)	128 (84.2)	24 (15.8)
Alcohol consumption						
Yes	30 (15.0)	28 (93.3)	2 (6.7)	121 (60.5)	98 (81.0)	23 (19.0)
No	170 (85.0)	145 (85.3)	25 (14.7)	79 (39.5)	65 (82.3)	14 (17.7)
Multivitamin use						
Yes	54 (27.0)	42 (77.8)	12 (22.2)	24 (12.0)	17 (70.8)	7 (29.2)
No	146 (73.0)	131 (89.7)	15 (10.3)	176 (88.0)	146 (83.0)	30 (17.0)
Calcium supplement use						
Yes	43 (21.5)	34 (79.1)	9 (20.9)	9 (4.5)	6 (66.7)	3 (33.3)
No	157 (78.5)	139 (88.5)	18 (11.5)	191 (95.5)	157 (82.2)	34 (17.8)
Folic acid use						
Yes	83 (41.5)	65 (78.3)	18 (21.7)	34 (17.0)	26 (76.5)	8 (23.5)
No	117 (58.5)	108 (92.3)	9 (7.7)	166 (83.0)	137 (82.5)	29 (17.5)
Milk intake frequency before pregnancy						
≥once a week	130 (65.0)	111 (85.4)	19 (14.6)	101 (50.5)	78 (77.2)	23 (22.8)
<once a week	70 (35.0)	62 (88.6)	8 (11.4)	99 (49.5)	85 (85.9)	14 (14.1)
Animal liver intake frequency before pregnancy						
≥once a week	40 (20.0)	36 (90.0)	4 (10.0)	36 (18.0)	32 (88.9)	4 (11.1)
<once a week	160 (80.0)	137 (85.6)	23 (14.4)	164 (82.0)	131 (79.9)	33 (20.1)
Deep-sea fish intake frequency before pregnancy						
≥once a week	109 (54.5)	96 (88.1)	13 (11.9)	99 (49.5)	82 (82.8)	17 (17.2)
<once a week	91 (45.5)	77 (84.6)	14 (15.4)	101 (50.5)	81 (80.2)	20 (19.8)
Season of blood sample collection						
Summer and autumn	109 (54.5)	88 (80.7)	21 (19.3)	109 (54.5)	79 (72.5)	30 (27.5)
Spring and winter	91 (45.5)	85 (93.4)	6 (6.6)	91 (45.5)	84 (92.3)	7 (7.7)
The total sperm count (×10^6^)				146 (81, 237)	144 (80, 232)	172 (88, 338)
Sperm concentration (×10^6^/mL)				50.9 (31.1, 88.0)	50.3 (31.2, 84.9)	59.3 (35.3, 105.8)
Progressive motile sperm count (×10^6^)				66 (35, 103)	62 (33, 100)	82 (47, 114)
The percentage of normal morphology sperm (%)				10.4 (8.3, 12.3)	10.3 (8.2, 12.1)	11.7 (9.3, 13.2)

**Table 2 nutrients-14-03058-t002:** Associations of serum 25(OH)D concentrations among preconception couples with clinical pregnancy and time to pregnancy.

Vitamin D	Clinical Pregnancy			Time to Pregnancy		
	cOR	Model 1 ^a^ aOR (95% CI)	Model 2 ^b^ aOR (95% CI)	cFR	Model 1 ^a^ aFR (95% CI)	Model 2 ^b^ aFR (95% CI)
Serum 25(OH)D concentrations among preconception women						
<30 ng/mL	Ref	Ref	Ref	Ref	Ref	Ref
≥30 ng/mL	0.92	1.02 (0.37~2.76)	0.90 (0.32~2.54)	1.08	1.07 (0.69~1.68)	1.08 (0.70~1.69)
Serum 25(OH)D concentrations among preconception men						
<30 ng/mL	Ref	Ref	Ref	Ref	Ref	Ref
≥30 ng/mL	3.15	3.29 (1.07~10.12)	3.72 (1.16~11.86)	1.32	1.37 (0.93~2.00)	1.50 (1.01~2.23)

Abbreviations: cOR, crude odds ratio; aOR, adjusted odds ratio; CI, confidence interval; cFR, crude fecundability ratio; aFR, adjusted fecundability ratio. ^a^ Model 1: data were adjusted for preconception age, preconception BMI, education, smoking, alcohol consumption, calcium supplementation, folic acid supplementation, and frequency of milk intake before pregnancy. ^b^ Model 2: data were adjusted for all covariates in Model 1 plus gravidity, multivitamin supplementation, and season of blood sample collection.

**Table 3 nutrients-14-03058-t003:** The associations of serum vitamin D levels during pregnancy and pregnancy outcomes (*n* = 198).

Characteristics	In the 2nd Trimester, *n* (%)/Mean ± SD/M (P_25_, P_75_)			In the 3rd Trimester, *n* (%)/Mean ± SD/M (P_25_, P_75_)		
	Overall	25(OH)D Groups		Overall	25(OH)D Groups	
		<30 ng/mL	≥30 ng/mL		<30 ng/mL	≥30 ng/mL
25(OH)D concentrations (ng/mL)	21.64 (17.67, 26.34)	170 (85.9)	28 (14.1)	19.08 (13.76, 25.49)	172 (86.9)	26 (13.1)
Multivitamin use during pregnancy						
Yes	146 (73.7)	121 (82.9)	25 (17.1)	146 (73.7)	124 (84.9)	22 (15.1)
No	52 (26.3)	49 (94.2)	3 (5.8)	52 (26.3)	48 (92.3)	4 (7.7)
Calcium supplement use during the pregnancy						
Yes	159 (80.3)	134 (84.3)	25 (15.7)	159 (80.3)	134 (84.3)	25 (15.7)
No	39 (19.7)	36 (92.3)	3 (7.7)	39 (19.7)	38 (97.4)	1 (2.6)
Folic acid use during pregnancy						
Yes	144 (72.7)	118 (81.3)	26 (18.1)	144 (72.7)	124 (86.1)	20 (13.9)
No	54 (27.3)	52 (96.3)	2 (3.7)	54 (27.3)	48 (88.9)	6 (11.1)
Weight gain during pregnancy						
Inadequate	59 (29.8)	50 (84.7)	9 (15.3)	59 (29.8)	50 (84.7)	9 (15.3)
Appropriate	86 (43.4)	72 (83.7)	14 (16.3)	86 (43.4)	73 (84.9)	13 (15.1)
Overmuch	53 (26.8)	48 (90.6)	5 (9.4)	53 (26.8)	49 (92.5)	4 (7.5)
Milk intake frequency during pregnancy						
≥once a week	134 (67.7)	112 (83.6)	22 (16.4)	134 (67.7)	115 (85.8)	19 (14.2)
<once a week	64 (32.3)	58 (90.6)	6 (9.4)	64 (32.3)	57 (89.1)	7 (10.9)
Animal liver intake frequency during pregnancy						
≥once a week	47 (23.7)	42 (89.4)	5 (10.6)	47 (23.7)	35 (74.5)	12 (25.5)
<once a week	151 (76.3)	128 (84.8)	23 (15.2)	151 (76.3)	137 (90.7)	14 (9.3)
Deep-sea fish food intake frequency during pregnancy						
≥once a week	104 (52.5)	90 (86.5)	14 (13.5)	104 (52.5)	86 (82.7)	18 (17.3)
<once a week	94 (47.5)	80 (85.1)	14 (14.9)	94 (47.5)	86 (91.5)	8 (8.5)
Gestational diabetes mellitus						
Yes	24 (12.1)	21 (12.4)	3 (10.7)	24 (12.1)	18 (10.5)	6 (23.1)
No	174 (87.9)	149 (87.6)	25 (89.3)	174 (87.9)	154 (89.5)	20 (76.9)
Gestational hypertension						
Yes	9 (4.5)	7 (4.1)	2 (7.1)	9 (4.5)	8 (4.7)	1 (3.8)
No	189 (95.5)	163 (95.9)	26 (92.9)	189 (95.5)	164 (95.3)	25 (96.2)
Gestational anemia						
Yes	61 (30.8)	52 (30.6)	9 (32.1)	61 (30.8)	58 (33.7)	3 (4.9)
No	137 (69.2)	118 (69.4)	19 (87.9)	137 (69.2)	114 (66.3)	23 (95.1)
Premature rupture of membranes						
Yes	21 (10.6)	20 (11.8)	1 (3.6)	21 (10.6)	20 (11.6)	1 (3.8)
No	177 (89.4)	150 (88.2)	27 (96.4)	177 (89.4)	152 (88.4)	25 (96.2)
Delivery mode						
Cesarean delivery	60 (30.3)	51 (30.0)	9 (32.1)	60 (30.3)	53 (30.8)	7 (26.9)
Vaginal delivery	138 (69.7)	119 (70.0)	19 (67.9)	138 (69.7)	119 (69.2)	19 (73.1)
Delivery gestational age	39.7 (38.8, 40.3)	39.7 (38.9, 40.3)	39.7 (38.9, 40.3)	39.7 (38.8, 40.3)	39.6 (38.9, 40.3)	39.9 (39.3,40.5)
Birth weight	3304 ± 405	3316 ± 407	3236 ± 391	3304 ± 405	3290 (±410)	3401 (±362)
Ponderal index	2.68 ± 0.22	2.68 ± 0.22	2.66 ± 0.2	2.68 ± 0.22	2.67 ± 0.21	2.76 ± 0.21

**Table 4 nutrients-14-03058-t004:** Associations of preconception and pregnancy vitamin D levels with pregnancy outcomes.

Vitamin D	GDM *	Gestational Anemia *	PROM *	Delivery Gestational Age *	PI *
	aOR (95% CI)	aOR (95%CI)	aOR (95% CI)	β (95% CI)	β (95% CI)
25(OH)D levels before pregnancy					
<30 ng/mL	Ref	Ref	Ref	Ref	Ref
≥30 ng/mL	2.23 (0.60~8.37)	1.23 (0.47~3.21)	0.89 (0.17~4.67)	−0.06 (−0.54~0.41)	0.01 (−0.08~0.10)
25(OH)D levels during the second trimester of the pregnancy					
<30 ng/mL	Ref	Ref	Ref	Ref	Ref
≥30 ng/mL	0.87 (0.21~3.55)	1.28 (0.51~3.23)	0.25 (0.03~2.10)	−0.07 (−0.54~0.40)	−0.01 (−0.10~0.08)
25(OH)D levels during the third trimester of the pregnancy					
<30 ng/mL	Ref	Ref	Ref	Ref	Ref
≥30 ng/mL	2.26 (0.72~8.88)	0.22 (0.06~0.82)	0.22 (0.02~1.99)	0.53 (0.05~1.01)	0.10 (0.01~0.19)

Abbreviations: aOR, adjusted odds ratio; CI, confidence interval; GDM, gestational diabetes mellitus; PROM, premature rupture of membranes; PI, ponderal index. * Adjusted for women’s age and BMI, education, household annual income per capita, gravidity, weight gain during pregnancy, multivitamin supplementation, calcium supplementation and folic acid, and consuming frequent milk and deep-sea fish food during pregnancy.

## Data Availability

Additional data are available from the corresponding author on reasonable request.

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
