# Peer review of "Serum Vitamin D Concentrations, Time to Pregnancy, and Pregnancy Outcomes among Preconception Couples: A Cohort Study in Shanghai, China"

_nutrients, 2022, doi:10.3390/nu14153058_

Round 1

Reviewer 1 Report

Dear authors,

great job, I find the topic interesting, the methodology nice and well presented. 

I would like to suggest to minor revisions

1) please mention within the introduction or the discussion the existence of a transient osteoporosis of the hip in which a low vitamine d and calcium levels play a role, a condition whom symptoms are often underestimated by Obstetrics an gyn that consider it only some neurological disorder.  PMID: 33989941.

2) I would add a summary of findings table always very useful for the reader to get the key points

otherwise I would recommend it for publication

Author Response

  1. please mention within the introduction or the discussion the existence of a transient osteoporosis of the hip in which a low vitamine d and calcium levels play a role, a condition whom symptoms are often underestimated by Obstetrics an gyn that consider it only some neurological disorder.  PMID: 33989941.

Response 1: We are grateful for the suggestion. We have modified the introduction to demonstrate the role of vitamin D deficiency in pregnancy outcomes including pre-eclampsia, gestational diabetes mellitus, gestational anemia, preterm birth, low birth weight, and cesarean section as follows.We have added 2 references [20, 21] as suggested to demonstrate the effect of vitamin D deficiency on cesarean section. Please see our revisions as the following.

Introduction paragraph 2: “Vitamin D deficiency and insufficiency has been recognized as a global public health problem that occurs in children, adolescents, pregnant women and the elderly worldwide [12]. Although there are many studies investigating the role of vitamin D during pregnancy in pregnancy outcomes, the findings are inconsistent [13-15]. Most studies have reported low gestational serum 25-hydroxyvitamin D (25(OH)D) linked to an increased risk of pre-eclampsia, gestational diabetes mellitus (GDM), gestational anemia, preterm birth (PTB) and low birth weight (LBW) [16-19]. Some studies suggested vitamin D deficiency during pregnancy was associated with cesarean section[20, 21]. However, very few studies observed the impact of preconception 25(OH)D concentrations on pregnancy outcomes.” (Page 2)

References:

  1. Palacios, C.; Gonzalez, L. Is vitamin D deficiency a major global public health problem? J Steroid Biochem Mol Biol. 2014, 144 Pt A, 138-145.
  2. Perez-Lopez, F.R.; Pasupuleti, V.; Mezones-Holguin, E.; Benites-Zapata, V.A.; Thota, P.; Deshpande, A.; Her-nandez, A.V. Effect of vitamin D supplementation during pregnancy on maternal and neonatal outcomes: a systematic review and meta-analysis of randomized controlled trials. Fertil. Steril. 2015, 103, 1278-1288.
  3. Agarwal, S.; Kovilam, O.; Agrawal, D.K. Vitamin D and its impact on maternal-fetal outcomes in pregnancy: A critical review. Crit Rev Food Sci Nutr. 2018, 58, 755-769.
  4. Roth, D.E.; Leung, M.; Mesfin, E.; Qamar, H.; Watterworth, J.; Papp, E. Vitamin D supplementation during pregnancy: state of the evidence from a systematic review of randomised trials. BMJ. 2017, 359, j5237.
  5. Baca, K.M.; Simhan, H.N.; Platt, R.W.; Bodnar, L.M. Low maternal 25-hydroxyvitamin D concentration in-creases the risk of severe and mild preeclampsia. Ann. Epidemiol. 2016, 26, 853-857.
  6. Arnold, D.L.; Enquobahrie, D.A.; Qiu, C.; Huang, J.; Grote, N.; VanderStoep, A.; Williams, M.A. Early preg-nancy maternal vitamin D concentrations and risk of gestational diabetes mellitus. Paediatr Perinat Epidemiol. 2015, 29, 200-210.
  7. Thomas, C.E.; Guillet, R.; Queenan, R.A.; Cooper, E.M.; Kent, T.R.; Pressman, E.K.; Vermeylen, F.M.; Rob-erson, M.S.; O'Brien, K.O. Vitamin D status is inversely associated with anemia and serum erythropoietin during pregnancy. Am. J. Clin. Nutr. 2015, 102, 1088-1095.
  8. Eremkina, A.K.; Mokrysheva, N.G.; Pigarova, E.A.; Mirnaya, S.S. Vitamin D: effects on pregnancy, maternal, fetal and postnatal outcomes. Ter Arkh. 2018, 90, 115-127.
  9. Quaresima, P.; Angeletti, M.; Luziatelli, D.; Luziatelli, S.; Venturella, R.; Di Carlo, C.; Bernardo, S. Pregnancy associated transient osteoporosis of the hip (PR-TOH): A non-obstetric indication to caesarean section. A case report with literature review. Eur J Obstet Gynecol Reprod Biol. 2021, 262, 28-35.
  10. Merewood, A.; Mehta, S.D.; Chen, T.C.; Bauchner, H.; Holick, M.F. Association between vitamin D deficiency and primary cesarean section. J Clin Endocrinol Metab. 2009, 94, 940-945.
  11. I would add a summary of findings table always very useful for the reader to get the key points otherwise I would recommend it for publication.

Response 2: Thanks for the suggestion. We have established a summary table of findings and plan to add it as a supplementary material in the manuscript. Please see the supplementary table “Associations of preconception and pregnancy vitamin D levels with clinical pregnancy, time to pregnancy and pregnancy outcomes” as follows.We have added one sentence at the end of Results section.

Please see our revisions as the following.

“The summary of the findings was shown in Supplementary table 1.”(page 7, last paragraph, Results)

Supplementary Table 1: Associations of preconception and pregnancy vitamin D levels with clinical pregnancy, time to pregnancy and pregnancy outcomes

Outcomes

Sufficient 25(OH)D levels (≥30 ng/ml) vs

insufficient 25(OH)D levels (<30 ng/ml)

25(OH)D levels among preconception women

25(OH)D levels among preconception men

25(OH)D levels during the 2nd trimester of the pregnancy

25(OH)D levels during the 3rd trimester of the pregnancy

Clinical pregnancy within six months

cOR(95%CI)

0.92(0.36~2.31)

3.15(1.06~9.39)*

--

--

aOR(95%CI)a

0.90(0.32~2.54)

3.72(1.16~11.86)*

--

--

Time to pregnancy

cFR(95%CI)

1.08(0.71~1.62)

1.32(0.92~1.90)

--

--

aFR(95%CI)a

1.08(0.70~1.69)

1.50(1.01~2.23)

--

--

Gestational diabetes mellitus

cOR(95%CI)

1.38(0.43~4.42)

--

0.85(0.24~3.07)

2.57(0.91~7.22)

aOR(95%CI)b

2.23(0.60~8.37)

--

0.87(0.21~3.55)

2.26(0.72~8.88)

Gestational anemia

cOR(95%CI)

0.99(0.41~2.44)

--

1.08(0.46~2.53)

0.26(0.07~0.89)*

aOR(95%CI)b

1.23(0.47~3.21)

--

1.28(0.51~3.23)

0.22(0.06~0.82)*

Premature rupture of membranes

cOR(95%CI)

0.67(0.15~3.07)

--

0.28(0.04~2.16)

0.30(0.04~2.37)

aOR(95%CI)b

0.89(0.17~4.67)

--

0.25(0.03~2.10)

0.22(0.02~1.99)

Delivery gestational age

β(95%CI)b

-0.06(-0.54~0.41)

--

-0.07(-0.54~0.40)

0.53(0.05~1.01)*

Ponderal index

β(95%CI)b

0.01(-0.08~0.10)

--

-0.01(-0.10~0.08)

0.10(0.01~0.19)*

Abbreviations: cOR, crude odds ratio; aOR, adjusted odds ratio; CI, confidence interval; cFR, crude fecundability ratio; aFR, adjusted fecundability ratio.

aAdjusted for preconception age, preconception BMI, education, smoking, alcohol consumption, gravidity, multivitamin supplementation, calcium supplementation, folic acid supplementation, taking frequent milk intake before pregnancy, and season of blood sample collection.

bAdjusted for women’s age and BMI, education, household annual income per capita, gravidity, weight gain during pregnancy, multivitamin supplementation, calcium supplementation and folic acid, and taking frequent milk and deep-sea fish foods during pregnancy.

*P<0.05.

Reviewer 2 Report

The authors divide their results into two groups: individuals with vitamin D levels below 30 ng/ml and those with vitamin D levels equal to or above 30 ng/ml (see Table 1). They use the dividing line of 30 ng/ml because it is, according to the Endocrine Society guidelines, the minimum level for vitamin D sufficiency. Finally, the authors use their results to argue that levels above 30 ng/ml are better for reproductive health.

The authors are assuming what they wish to prove. If the dividing line had been 20 ng/ml, they would have probably found that levels above 20 ng/ml are associated with better reproductive health than levels below 20 ng/ml. The choice of 30 ng/ml is arbitrary. That choice would have made sense if there had been roughly equal numbers of participants below and above that dividing line. In reality, 173 participants were below 30 ng/ml and 27 were above among the women, and 163 were below and 37 above among the men.

The authors should have divided their results into four categories:

Below 10 ng/ml

10 ng/ml to 19 ng/ml

20 ng/ml to 29 ng/ml

30 ng/ml and above

The authors could have then charted the relationship between vitamin D level and reproductive success. Does reproductive success progressively increase up to 20 ng/ml with no further increases at higher levels?  Or is there no limit? Does reproductive success progressively increase with progressively higher vitamin D levels up to and beyond 30 ng/ml? These questions should be addressed, and the authors missed an opportunity to answer them.

The authors may reply that the Endocrine Society guidelines should apply to reproductive health because they apply to other aspects of human health. Again, they would be assuming what they wish to prove. It is also debatable whether guidelines derived from studies of North American participants are applicable to participants in other human populations.

Author Response

  1. The authors are assuming what they wish to prove. If the dividing line had been 20 ng/ml, they would have probably found that levels above 20 ng/ml are associated with better reproductive health than levels below 20 ng/ml. The choice of 30 ng/ml is arbitrary. That choice would have made sense if there had been roughly equal numbers of participants below and above that dividing line. In reality, 173 participants were below 30 ng/ml and 27 were above among the women, and 163 were below and 37 above among the men.

The authors should have divided their results into four categories:

Below 10 ng/ml

10 ng/ml to 19 ng/ml

20 ng/ml to 29 ng/ml

30 ng/ml and above

The authors could have then charted the relationship between vitamin D level and reproductive success. Does reproductive success progressively increase up to 20 ng/ml with no further increases at higher levels?  Or is there no limit? Does reproductive success progressively increase with progressively higher vitamin D levels up to and beyond 30 ng/ml? These questions should be addressed, and the authors missed an opportunity to answer them.

The authors may reply that the Endocrine Society guidelines should apply to reproductive health because they apply to other aspects of human health. Again, they would be assuming what they wish to prove. It is also debatable whether guidelines derived from studies of North American participants are applicable to participants in other human populations.

Response: Thanks for the suggestion. We have conducted the analysis according to the suggestion. Since no man had a vitamin D level below 10 ng/ml, only 1 woman had vitamin D level below 10 ng/ml, 1 pregnant woman had vitamin D level below 10 ng/ml in the second trimester and 14 pregnant woman had vitamin D level below 10 ng/ml in the third trimester, we would like to divide the vitamin D level into three categories:Below 20 ng/ml, 20 ng/ml to 29 ng/ml, 30 ng/ml and above (table 1). The results are shown in the table 2“Associations of serum 25(OH)D concentrations among preconception couples with clinical pregnancy and time to pregnancy ” and table 3 “Associations of preconception and pregnancy vitamin D levels with pregnancy outcomes” as follows.

As shown by the table 2, the couples had higher conception rates within six months (adjusted odds ratio(aOR): 4.15, 95% CI: 1.08, 16.00) and reduced time to pregnancy (adjusted fecundability ratio (aFR): 1.36, 95%CI: 1.03, 1.79) if male partners had sufficient 25(OH)D (≥30 ng/ml) compared with deficient 25(OH)D (<20 ng/ml). This was similar to the results shown in the manuscript when we divided vitamin D level into two categories: Below 30 ng/ml, 30 ng/ml and above. As shown by the table 3, when compared to pregnant women with vitamin D deficient 25(OH)D (<20 ng/ml) in the 3rd trimester of pregnancy, sufficient 25(OH)D(≥30 ng/ml) was associated with reduced odds of anemia (OR:0.25, 95%CI: 0.06, 0.96), longer gestational age (β: 0.58, 95%CI: 0.08, 1.08) and newborns’ higher ponderal index (β: 0.12, 95%CI: 0.03, 0.22), which was consistent with the contents in our original manuscript.

Since the non-statistical difference between the groups of 25(OH)D level of 20~30ng/ml and below20 ng/ml might be due to the the inadequate statistic power, we would like to use the 30 ng/ml as the cutoff point for the analysis. This cutoff point was also used in a study [28] which was referenced in our manuscript. However, we have included the only use of cutoff point of 30 ng/ml for analysis as a limitation in the discussion.

Please see our revisions as the following.

 “Moreover, we used 30 ng/ml as sufficiency cutoff point according to the Endocrine Society guidelines [32]. However, it is needed to ascertain the associations between vitamin D and reproductive health outcomes using different cutoff points since there was a debate whether guidelines derived from studies of North American participants are applicable to participants in other human populations such as Asian population. ” (Page 11, paragraph 8, Discussion)

References:

  1. 28. Mumford, S.L.; Garbose, R.A.; Kim, K.; Kissell, K.; Kuhr, D.L.; Omosigho, U.R.; Perkins, N.J.; Galai, N.; Silver, R.M.; Sjaarda, L.A.; Plowden, T.C.; Schisterman, E.F. Association of preconception serum 25-hydroxyvitamin D concentrations with livebirth and pregnancy loss: a prospective cohort study. The Lancet Diabetes & Endo-crinology. 2018, 6, 725-732.
  2. 32. Holick, M.F.; Binkley, N.C.; Bischoff-Ferrari, H.A.; Gordon, C.M.; Hanley, D.A.; Heaney, R.P.; Murad, M.H.; Weaver, C.M. Evaluation, treatment, and prevention of vitamin D deficiency: an Endocrine Society clinical practice guideline. J Clin Endocrinol Metab. 2011, 96, 1911-1930.

Table 1: Distribution of 25(OH)D levels among the participants

25(OH)D groups

Preconception women, n(%)

Preconception men, n(%)

Pregnant women in the 2nd trimester, n(%)

Pregnant women in the 3rd trimester, n(%)

<20 ng/ml

62(31.0)

39(19.5)

84(42.4)

106(53.5)

20 ng/ml~30 ng/ml

111(55.5)

124(62.0)

86(43.4)

66(33.3)

≥30 ng/ml

27(13.5)

37(18.5)

28(14.2)

26(13.2)

Table2: Associations of serum 25(OH)D concentrations among preconception couples with clinical pregnancy and time to pregnancy

Vitamin D

Clinical pregnancy

Time to pregnancy

cOR

Model 1a aOR(95%CI)

Model 2b aOR(95%CI)

cFR

Model 1a aFR(95%CI)

Model 2b aFR(95%CI)

Serum 25(OH)D concentrations among preconception women

<20 ng/ml

Ref

Ref

Ref

Ref

Ref

Ref

20~30 ng/ml

0.56(0.25~1.21)

0.54(0.24~1.19)

0.53(0.24~1.18)

0.93(0.76~1.13)

0.93(0.75~1.15)

0.92(0.74~1.14)

≥30 ng/ml

0.62(0.21~1.81)

0.67(0.22~2.12)

0.64(0.19~2.12)

1.04(0.79~1.38)

1.04(0.78~1.40)

1.05(0.78~1.43)

Serum 25(OH)D concentrations among preconception men

<20 ng/ml

Ref

Ref

Ref

Ref

Ref

Ref

20~30 ng/ml

1.23(0.56~2.70)

1.10(0.47~2.61)

1.15(0.48~2.74)

1.19(0.82~1.73)

0.97(0.79~1.19)

0.95(0.78~1.17)

≥30 ng/ml

3.67(1.06~12.68)*

3.54(0.97~13.00)

4.15(1.08~16.00)*

1.51(0.95~2.40)

1.27(0.98~1.66)

1.36(1.03~1.79)*

 Abbreviations: cOR, crude odds ratio; aOR, adjusted odds ratio; CI, confidence interval; cFR, crude fecundability ratio; aFR, adjusted fecundability ratio.

aModel 1: data were adjusted for preconception age, preconception BMI, education, smoking, alcohol consumption, calcium supplementation, folic acid supplementation, and taking frequent milk intake before pregnancy.

bModel 2: data were adjusted for all covariates in Model 1 plus gravidity, multivitamin supplementation, and season of blood sample collection.

*P<0.05.

Table 3: Associations of preconception and pregnancy vitamin D levels with pregnancy outcomes

Vitamin D

GDMa

Gestational anemiaa

PROMa

Delivery gestational agea

PIa

aOR(95%CI)

aOR(95%CI)

aOR(95%CI)

β(95%CI)

β(95%CI)

25(OH)D levels before pregnancy

<20 ng/ml

Ref

Ref

Ref

Ref

Ref

20~30 ng/ml

1.01(0.38~3.02)

0.60(0.29~1.23)

0.43(0.15~1.26)

0.05(-0.30~0.10)

0.01(-0.06~0.08)

≥30 ng/ml

2.05(0.48~8.80)

0.82(0.29~2.33)

0.47(0.08~2.68)

-0.01(-0.53~0.51)

0.02(-0.08~0.12)

25(OH)D levels during the second trimester of the pregnancy

<20 ng/ml

Ref

Ref

Ref

Ref

Ref

20~30 ng/ml

0.74(0.27~2.04)

0.92(0.45~1.87)

2.12(0.73~6.13)

0.01(-0.33~0.36)

-0.04(-0.11~0.02)

≥30 ng/ml

0.71(0.16~3.05)

1.00(0.37~2.74)

0.37(0.04~3.46)

-0.04(-0.54~0.46)

-0.02(-0.12~0.07)

25(OH)D levels during the third trimester of the pregnancy

<20 ng/ml

Ref

Ref

Ref

Ref

Ref

20~30 ng/ml

0.96(0.31~3.00)

1.49(0.71~3.11)

0.73(0.23~2.26

0.27(-0.96~0.63)

0.04(-0.03~0.11)

≥30 ng/ml

2.52(0.69~9.17)

0.25(0.06~0.96)*

0.19(0.02~1.83)

0.58(0.08~1.08)*

0.12(0.03~0.22)*

Abbreviations: aOR, adjusted odds ratio; CI, confidence interval; GDM, gestational diabetes mellitus; PROM, Premature rupture of membranes; PI, Ponderal index.

aAdjusted for women’s age and BMI, education, household annual income per capita, gravidity, weight gain during pregnancy, multivitamin supplementation, calcium supplementation and folic acid, and taking frequent milk and deep-sea fish foods during pregnancy.

*P<0.05.
